# MetaCC allows scalable and integrative analyses of both long-read and short-read metagenomic Hi-C data

**Yuxuan Du** [1] **& Fengzhu Sun** [1] ✉

Metagenomic Hi-C (metaHi-C) can identify contig-to-contig relationships with respect to their proximity within the same physical cell. Shotgun libraries in metaHi-C experiments can be constructed by next-generation sequencing (short-read metaHi-C) or more recent third-generation sequencing (long-read metaHi-C). However, all existing metaHi-C analysis methods are developed and benchmarked on short-read metaHi-C datasets and there exists much room for improvement in terms of more scalable and stable analyses, especially for long-read metaHi-C data. Here we report MetaCC, an efficient and integrative framework for analyzing both short-read and long-read metaHi-C datasets. MetaCC outperforms existing methods on normalization and binning. In particular, the MetaCC normalization module, named NormCC, is more than 3000 times faster than the current state-of-the-art method HiCzin on a complex wastewater dataset. When applied to one sheep gut long-read metaHi-C dataset, MetaCC binning module can retrieve 709 high-quality genomes with the largest species diversity using one single sample, including an expansion of five uncultured members from the order *Erysipelotrichales*, and is the only binner that can recover the genome of one important species *Bacteroides vulgatus*. Further plasmid analyses reveal that MetaCC binning is able to capture multi-copy plasmids.

Metagenomics aims to study the complex community structures and reveal metabolic potentials in microbial ecosystems without the isolation or cultivation of microbes in the environment[1–4]. The recent introduction of high-throughput chromosome conformation capture technique (Hi-C) into metagenomics provides new insights into species diversity and interactions between microorganisms within a single microbial sample[5–10].

Metagenomic Hi-C technique (metaHi-C) combines the rapidly developed proximity ligation approach with the metagenomic shotgun sequencing. Specifically, shotgun experiments directly extract genomic fragments from a single microbial sample. In parallel, Hi-C experiments on the same microbial sample create DNA–DNA proximity ligations between loci within the same physical cell, generating millions of paired-end Hi-C short reads. Fragmented shotgun reads are assembled into contiguous sequences, termed contigs, to which paired-end Hi-C reads are subsequently aligned. Therefore, metagenomic Hi-C contacts, defined as the numbers of Hi-C read pairs linking any pair of assembled contigs, reflect the contig-to-contig relationships with respect to their proximity. Since raw metagenomic Hi-C contacts are substantially affected by systematic biases, normalization is necessary after processing raw metaHi-C data[11–13]. We previously disclosed three systematic biases including the number of enzymatic restriction sites on contigs, contig length, and contig coverage and put forward a state-of-the-art normalization method HiCzin that can correct all three biases[14]. After the normalization, fragmented contigs can be grouped into metagenome-assembled genomes (MAGs)[15] using Hi-C contacts. This process, termed Hi-C-based binning, enables the construction of large compendia of metagenomic assembled

[1]Department of Quantitative and Computational Biology, University of Southern California, Los Angeles, CA, USA. ✉e-mail: fsun@usc.edu

microbial genomes. Several Hi-C-based binning methods have been designed, such as MetaTOR[16], bin3C[17], and HiCBin[18].

Despite recent advances of these computational tools designed for metagenomic Hi-C data, there still exists much room for improvement for more scalable and stable analyses. For instance, the Knight-Ruiz algorithm[19] utilized by bin3C[17] may fail to generate a bistochastic matrix when the raw Hi-C contact matrix is highly sparse[20,21]. MetaTOR[16] employs the classical Newman-Girvan modularity function[22] in its binning procedure, which cannot identify small genomes due to the resolution limit[23] in complex Hi-C contact networks[18]. HiCzin[14] and HiCBin[18] require a large amount of computing resources on estimating contig abundances and generating contig annotations, which refers to assigning nucleotide sequences to various taxonomic levels. Specifically, HiCzin and HiCBin utilize TAXAassign[24] to label contigs at the species level by running BLAST[25] against a curated nucleotide reference database. Moreover, since shotgun libraries in the original metagenomic Hi-C experiments are constructed using next-generation sequencing (short-read metaHi-C)[10–13], all existing computational methods are designed and merely benchmarked on short-read metaHi-C datasets[16–18]. With the rapid development of third-generation sequencing, multiple recent metaHi-C experiments also leveraged Nanopore or PacBio sequencing to generate long-read shotgun libraries (long-read metaHi-C)[26–29]. However, the current computational tools that have achieved state-of-the-art results on short-read metaHi-C datasets encounter difficulties in adapting to long-read metaHi-C datasets. In our experiments for this study, we observe that the performances of HiCBin, which demonstrated the superior binning performance on short-read metaHi-C datasets according to recent benchmarking studies[30], are markedly deteriorated on long-read metaHi-C datasets. One essential factor contributing to this decline is the large degradation of HiCBin as well as its adopted normalization method HiCzin when only a small fraction of assembled contigs can be successfully labeled at the species level by TAXAassign (Supplementary Note 1). Additionally, the taxonomic labeling of contigs assembled from long reads poses a challenge for TAXAassign (Supplementary Table 1), consequently limiting the effectiveness of HiCBin and HiCzin on long-read metaHi-C datasets. Therefore, it is imperative to develop new computational methods to fill these gaps.

Here we report MetaCC, a scalable and integrative framework for both long-read and short-read metaHi-C datasets. In the MetaCC framework, raw metagenomic Hi-C contacts are first efficiently and effectively normalized by a new normalization method, NormCC. In comparison to HiCzin, which relies on estimated contig abundances as input, NormCC employs a negative binomial regression model to represent contig abundances based on easily obtainable features including the number of restriction sites on contigs, contig length, and the number of proximity ligation events within contigs. Consequently, NormCC does not require the estimation of contig abundances. Additionally, HiCzin models the Hi-C contacts between contigs of the same species, necessitating contig annotations. Conversely, NormCC models the total number of proximity ligation events for each contig using a second negative binomial regression, eliminating the need for contig annotation. Using a synthetic yeast dataset[11], we validate the normalization performance of NormCC and show that NormCC outperforms HiCzin with respect to the spurious contact (i.e., Hi-C contacts linking contigs from different genomes due to experimental noises) removal, contig clustering, and computational time. Leveraging NormCC-normalized Hi-C contacts, the binning module in MetaCC enables the retrieval of high-quality MAGs. We compare the retrieval performance of MetaCC binning against all publicly-available Hi-C-based binning tools MetaTOR, bin3C, and HiCBin as well as one state-of-the-art shotgun-based binner VAMB[31] on two real short-read metaHi-C datasets and two real long-read metaHi-C datasets. Downstream annotation studies and plasmid analyses on long-read metaHi-C datasets demonstrate the superior ability of MetaCC on characterizing the species diversity, extracting important microbes out of the microbial ecosystems, and capturing multi-copy plasmid contigs.

## Results

### Overview of MetaCC

MetaCC is a comprehensive analysis framework designed for both short-read and long-read metaHi-C datasets (Fig. 1a) and consists of four main components. (I) We design a scalable and effective normalization method, NormCC, to eliminate systematic biases from the raw metagenomic Hi-C contact matrix. (II) We discard spurious inter-species Hi-C contacts linking contigs from different species due to experimental noises. (III) Based on the normalized Hi-C contact graph, we retrieve high-quality MAGs using Leiden clustering[32] with all hyper-parameters automatically tuned. (IV) With several new computational strategies, we reliably characterize the structure of microbial ecosystems.

### NormCC comprehensively corrected all systematic biases existing in a synthetic yeast metaHi-C dataset

Leveraging a synthetic yeast metaHi-C dataset[11] with all assembled contigs labeled at the species level, we previously revealed that raw intra-species Hi-C contacts, defined as the number of proximity ligation events linking contigs from the same species, were more enriched between pairs of contigs with a larger number of restriction sites, longer contigs, and/or contigs with higher coverages[14]. We have also demonstrated that HiCzin, the normalization method employed in HiCBin, outperformed other metaHi-C-based normalization methods, including those utilized in bin3C and MetaTOR, in terms of spurious contact detection and contig clustering using the synthetic yeast metaHi-C dataset[14,18]. Notably, HiCzin incorporates contig annotations at the species level, obtained through TAXAassign[24], to select intra-species Hi-C contacts utilized in fitting its normalization model. In line with the previous analyses, we validated the performance of NormCC normalization on this synthetic sample and compared it to HiCzin using the same benchmarking criteria. Details of processing raw data were shown in the Methods section.

The procedures of NormCC normalization and spurious contact removal can be visualized in Fig. 1b. To quantify the biases existing for raw intra-species Hi-C contacts, we computed the Pearson correlation coefficients between all raw intra-species contacts and the product of the number of restriction sites, the length, and the coverage for corresponding contig pairs, which were 0.429, 0.400, and 0.184, respectively, indicating the strong biases of three factors on raw metagenomic Hi-C contacts. After the NormCC normalization, the correlation coefficients between the bias-corrected Hi-C contacts and the product of three factors were decreased to 0.094, 0.090, and 0.004, respectively, demonstrating that NormCC was able to comprehensively correct all systematic biases for the metaHi-C datasets.

### NormCC outperformed HiCzin on the spurious contact removal, contig clustering, and computational time

Though the magnitude of the spurious inter-species contacts is significantly smaller than that of the intra-species contacts in the NormCC-normalized Hi-C contact matrix (Supplementary Fig. 1), discarding all Hi-C contacts below a threshold as spurious inevitably resulted in the loss of a few informative intra-species contacts. Therefore, the improved capacity for removing spurious contacts from one single Hi-C contact matrix can be assessed by effectively eliminating a greater number of spurious contacts while minimizing the unintended removal of informative intra-species contacts. We then applied the spurious contact removal strategy (see Methods) based on the raw, HiCzin-normalized, or NormCC-normalized Hi-C contact matrices, respectively, and plotted discard-retain (DR) curves where the proportion of discarded spurious contacts among all spurious

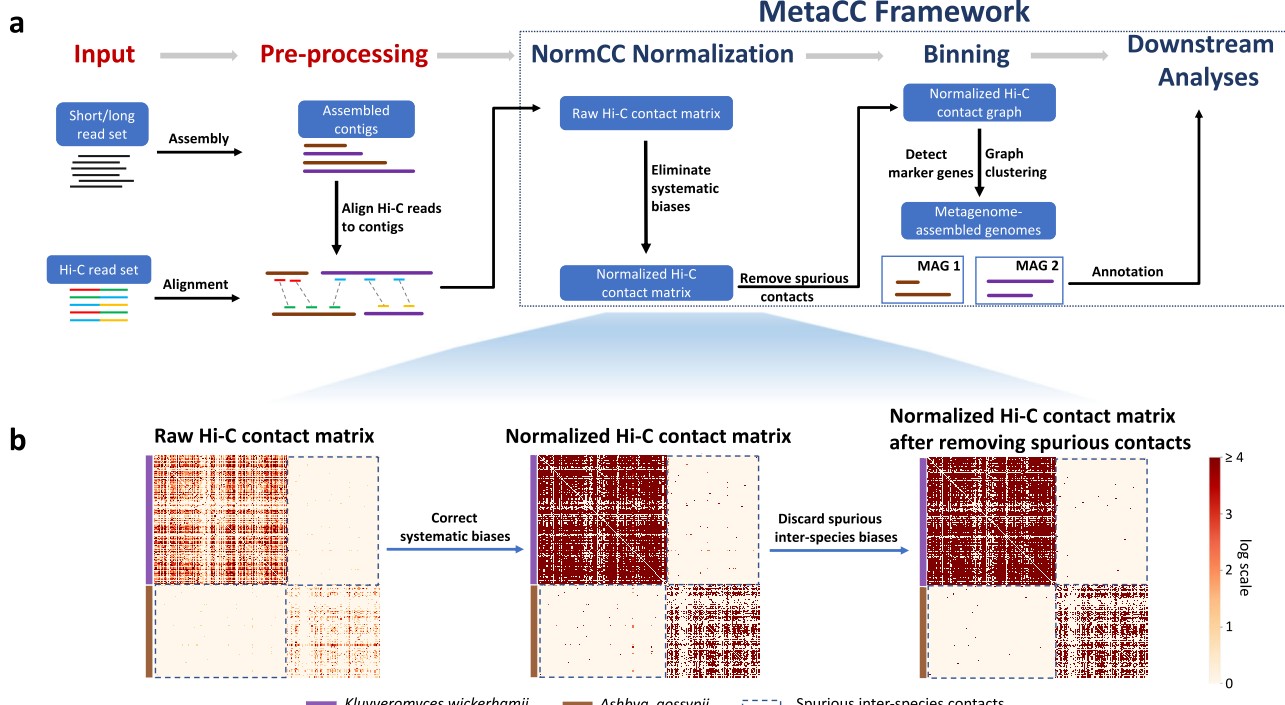

**Fig. 1 | Overview of the MetaCC framework for metagenomic Hi-C analyses.**
**a** The input metaHi-C dataset consists of shotgun libraries and Hi-C libraries. Short/long reads in shotgun libraries are assembled into contigs, to which Hi-C paired-end reads are subsequently aligned. In this way, raw Hi-C contact matrix displaying the proximity similarity between contigs within cells can be constructed. The raw Hi-C contact matrix is normalized by the NormCC normalization module to correct the systematic biases and spurious inter-species contacts are subsequently removed. Assembled contigs are then binned into high-quality MAGs leveraging the normalized Hi-C contact matrix. Finally, downstream analyses are conducted. **b** Visualize the procedures of NormCC normalization and spurious contact removal by plotting heatmaps of the Hi-C contact matrix for contigs belonging to the species *Kluyveromyces wickerhamii* and *Ashbya gossypii* from a synthetic yeast dataset.

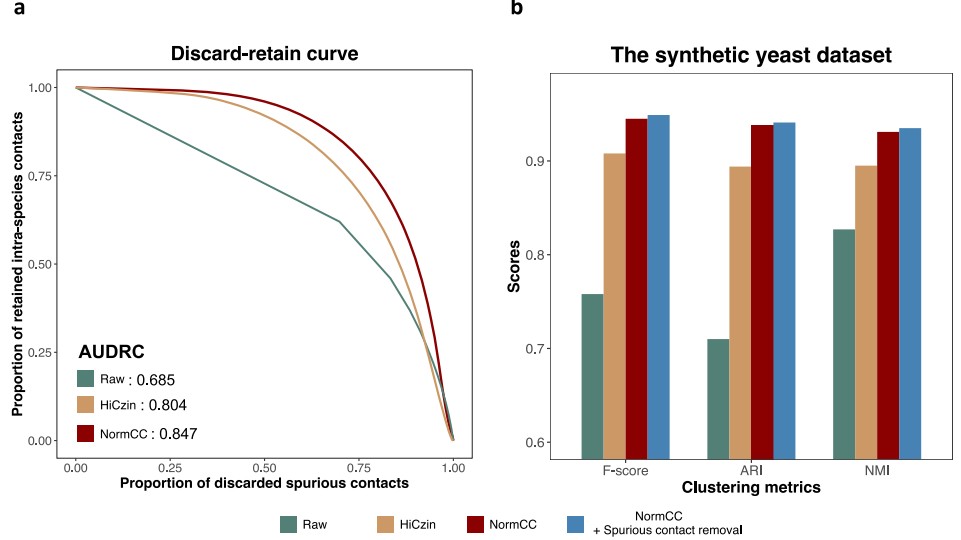

**Fig. 2 | Benchmarking the NormCC normalization module on the synthetic yeast metaHi-C dataset. a** Discard-retain curves for evaluating spurious contact removal based on the raw, HiCzin-normalized, or NormCC-normalized Hi-C contact matrices, respectively. NormCC achieved the highest AUDRC (i.e., area under discard-retain curve). **b** Performance of contig clustering based on the raw, HiCzin-normalized, or NormCC-normalized Hi-C contact matrices as well as NormCC-normalized Hi-C contact matrix with spurious contact removal, respectively. NormCC outperformed HiCzin on the contig clustering in terms of F-score, ARI, and NMI.

contacts is plotted against the proportion of retained intra-species contacts within all intra-species contacts at various thresholds corresponding to various percentiles (Fig. 2a). The area under the discard-retain curve (AUDRC) can measure the ability of spurious contact removal. With respect to AUDRC, NormCC outperformed HiCzin.

Moreover, we applied the Leiden clustering strategy (see Methods) on the raw, HiCzin-normalized, or NormCC-normalized Hi-C contact matrices, respectively, to cluster contigs. To explore the impact of spurious contact removal on contig clustering, we also grouped contigs based on the NormCC-normalized Hi-C contact

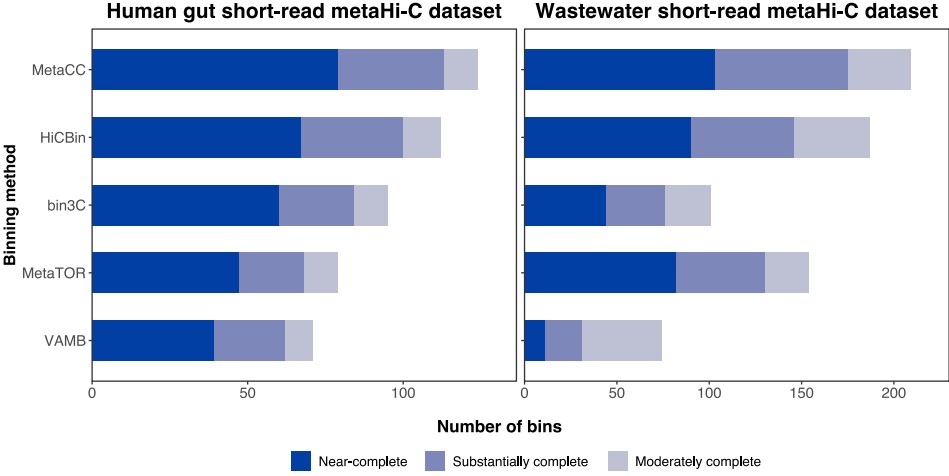

**Fig. 3 | Benchmarking the MetaCC binning module on short-read metaHi-C datasets.** MetaCC binning outperformed other binners on both the human gut and wastewater short-read metaHi-C datasets according to the CheckM criteria (Near-complete: completeness ≥ 90% and contamination ≤ 10%; Substantially complete: 70% ≤ completeness < 90% and contamination ≤ 10%; Moderately complete: 50% ≤ completeness < 70% and contamination ≤ 10%).

matrix after removing spurious contacts. Since the true species identity of all contigs was available, we employed three comprehensive metrics (Supplementary Note 2): Fowlkes-Mallows score (F-score), Adjusted Rand Index (ARI), and normalized mutual information (NMI) to evaluate the clustering performance. As shown in Fig. 2b, the bias elimination and the spurious contact removal could improve the clustering performance while NormCC outperformed HiCzin on the contig binning in terms of F-score, ARI, and NMI.

Finally, NormCC and HiCzin were executed on a 2.40 GHz Intel Xeon Processor E5-2665 with 50,000 MB RAM provided by the Advanced Research Computing platform at the University of Southern California. The time recording started at the input of raw Hi-C contact matrix and ended at the output of normalized Hi-C contact matrix. We ran both NormCC and HiCzin on the synthetic yeast dataset as well as four real metaHi-C datasets. The results of running time were shown in Supplementary Table 2. NormCC is much faster than HiCzin on all datasets. In particular, NormCC is more than 3000 × faster than HiCzin on the wastewater short-read metaHi-C dataset[10]. Apart from the running time, HiCzin consumed a large amount of extra computational resources to prepare the input data, including generating contig annotations and estimating the contig abundances, compared to NormCC.

## MetaCC binning achieved the best performance of MAG retrieval on short-read metaHi-C datasets
To validate MetaCC binning on short-read metaHi-C datasets, we applied it to two datasets from different microbial environments: human gut[13] and wastewater[10]. Since the actual genomes are unknown in real samples, we leveraged CheckM[33] to evaluate the quality of the recovered bins (see Methods). We compared MetaCC binning to other three publicly-available Hi-C-based binning tools, i.e., MetaTOR[16], bin3C[17], and HiCBin[18]. Additionally, we included one state-of-the-art shotgun-based binning method VAMB[31] into comparison. Without using Hi-C information, the shotgun-based binning depends on the sequence similarity and abundance features of contigs to retrieve draft genomic bins. In both datasets, MetaCC binning recovered more near-complete and high-quality bins than the alternatives considered (Fig. 3). Specifically, on the human gut dataset, VAMB as well as three Hi-C-based binning methods, i.e., MetaTOR, bin3C, and HiCBin, could recover 39, 47, 60, and 67 near-complete MAGs, respectively, while MetaCC binning increased this number to 79. Moreover, VAMB, MetaTOR, bin3C, and HiCBin retrieved 11, 82, 44, and 94 near-complete MAGs, respectively, which was improved to 103 by MetaCC

binning on the wastewater dataset. Notably, in all instances, Hi-C-based binning pipelines outperformed the shotgun-based method on short-read metaHi-C datasets, indicating the great potential of Hi-C information.

Additionally, for the human gut short-read metaHi-C dataset, we assessed the number of bins corresponding to known bacteria identified in the human gut environment and the number of bins that might contain chimeric genomes for different binning methods using UHGG gut microbial reference database[34] (see Methods). MetaCC binning recovered the largest number of known bacteria from the human gut environment based on the UHGG database. Specifically, VAMB, Meta-TOR, bin3C, HiCBin, and MetaCC binning retrieved 83, 107, 89, 118, and 128 bins, respectively, which were assigned to only one known species. Furthermore, only one bin was detected as chimeric for MetaCC binning, while 4, 6, 2, and 11 chimeric bins were identified for VAMB, MetaTOR, bin3C, and HiCBin, respectively.

## MetaCC binning markedly outperformed existing binners on long-read metaHi-C datasets
Since all previous studies only compared Hi-C-based binning tools on short-read metaHi-C datasets, we focused on the benchmarking of MetaCC binning and other existing Hi-C-based binners on long-read metaHi-C datasets leveraging one cow rumen long-read metaHi-C dataset[26] and one sheep gut long-read metaHi-C dataset[28]. Results were shown in Fig. 4a. On the cow rumen long-read metaHi-C dataset, VAMB, MetaTOR, and bin3C created 4, 5, and 5 near-complete MAGs, respectively, while MetaCC binning increased this number to 8. In total, MetaCC binning reconstructed 71 high-quality bins, a gain of 38 (115%), 28 (65.1%) and 31 (77.5%) high-quality bins against VAMB, MetaTOR and bin3C, respectively. HiCBin failed to bin contigs on the cow rumen dataset due to the nonconvergence of its adopted normalization method HiCzin. As for the sheep gut long-read metaHi-C dataset, VAMB generated 190 near-complete, 94 substantially complete, and 94 moderately complete bins. MetaTOR created 228 near-complete, 102 substantially complete, and 105 moderately complete MAGs. bin3C recovered 268 near-complete, 83 substantially complete, and 51 moderately complete draft genomic bins. HiCBin reconstructed 99 near-complete, 55 substantially complete, and 54 moderately complete bins. In contrast, MetaCC binning retrieved 417 near-complete, 162 substantially complete, and 130 moderately complete MAGs, significantly outperforming VAMB, MetaTOR, bin3C, and HiCBin with an increase of 227 (119.5%), 189 (82.9%), 149 (55.6%), and 318 (321%) near-complete bins, respectively. MetaCC binning also improved the

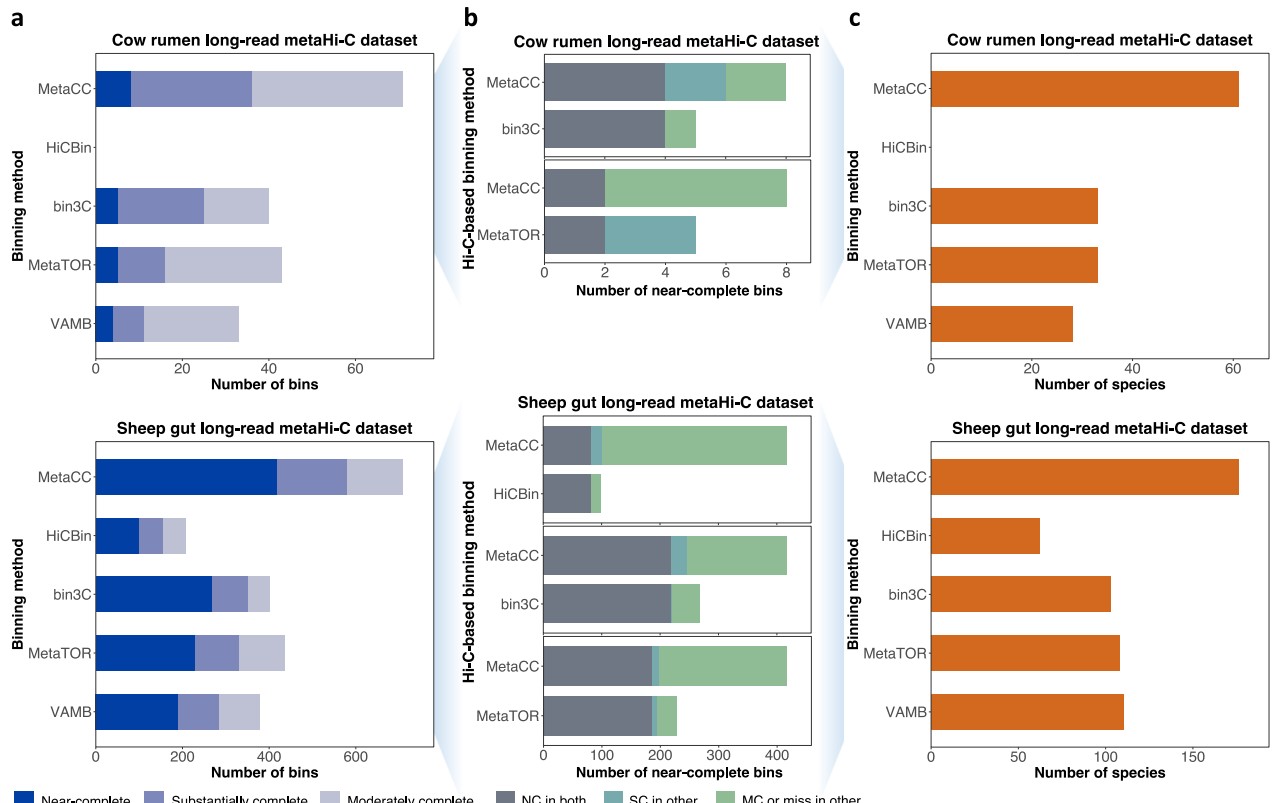

**Fig. 4 | Benchmarking the MetaCC binning module on long-read metaHi-C datasets. a** MetaCC binning outperformed other binners on both the cow rumen and sheep gut long-read metaHi-C datasets according to the CheckM criteria (Near-complete: completeness ≥ 90% and contamination ≤ 10%; Substantially complete: 70% ≤ completeness < 90% and contamination ≤ 10%; Moderately complete: 50% ≤ completeness <70% and contamination ≤ 10%). HiCBin failed to bin contigs on the cow rumen dataset due to the nonconvergence of its adopted normalization method HiCzin. **b** Comparison of near-complete bins identified by MetaCC binning and other Hi-C-based binners from the long-read metaHi-C datasets. The total length of each bar shows the total number of near-complete (NC) bins recovered by each binner. Each bar is then colored according to the number of NC bins that can be identified by both binners (NC in both), the number of NC bins that are substantially complete in the other bin set (SC in other), and the number of NC bins that are moderately complete or missing in the other bin set (MC or miss in other). **c** Comparison of the number of species recovered by different binners with high quality. MAGs retrieved by MetaCC binning represent the largest taxonomic diversity at the species level.

total number of high-quality MAGs by 331 (87.6%), 274 (63.0%), 307 (76.4%), and 501 (240.9%) compared to VAMB, MetaTOR, bin3C, and HiCBin, respectively. We also tested the efficacy of polishing HiFi assemblies using short reads on binning and found it did not improve the binning performance on the sheep gut dataset (Supplementary Note 3), suggesting that the polishing step might not be necessary and could be omitted in the future possibly due to the high accuracy of HiFi reads.

Moreover, we used Mash[35] to identify instances that MetaCC binning and other Hi-C-based binners (i.e., MetaTOR, bin3C, and HiC-Bin) retrieved the same near-complete MAGs on both long-read metaHi-C datasets. As shown in Fig. 4b, most of near-complete MAGs recovered by MetaTOR, bin3C, and HiCBin could also be retrieved by MetaCC binning in near-complete quality. MetaCC binning further reconstructed a large number of near-complete MAGs that were only recovered in substantially and moderately complete quality (or absent) by other Hi-C-based binners on both long-read metaHi-C datasets and the inverse cases were relatively rare, validating the superior ability of MetaCC binning to retrieve near-complete bins on long-read metaHi-C datasets.

Finally, we explored the capability of different binners to capture the species diversity in microbial samples by annotating all high-quality bins generated by MetaCC and other Hi-C-based binners on both long-read metaHi-C datasets using GTDB-TK[36]. As shown in Fig. 4c, bins derived from MetaCC binning represented a larger taxonomic diversity at the species level on both datasets. Additionally, we

found that one near-complete MAG (BIN 1254; Completeness: 97.94 and Contamination: 0.38) retrieved by MetaCC binning from the sheep gut samples belonged to one species *Bacteroides vulgatus*, which is one of the most important species in gut environments and plays important roles in inhibiting atherosclerosis and decreasing the production of the gut microbial lipopolysaccharide[37]. However, this important species could not be recovered by other binners with high quality from the sheep gut dataset. Therefore, MetaCC binning outperformed other binners on extracting the species structure out of microbial ecosystems.

## MetaCC binning identified and expanded the order *Erysipelotrichales* from the cow rumen and sheep gut samples

Members of the order *Erysipelotrichales*, which are found to have very important functions in animal disease and physiology[38], have been isolated from the human[38], cow[39], insect[40], and mouse[41] gut. Among high-quality MAG sets recovered by different binners, we found that only the set of MetaCC binning included the draft genome from the order *Erysipelotrichales* on the cow rumen dataset.

Similar to other gut environments, we also observed the prevalence of this order from the sheep gut sample, indicating an increasingly important role of the order *Erysipelotrichales* in animal microbiomes. Specifically, according to the annotation results of GTDB-TK, eight high-quality MAGs retrieved by MetaCC binning belonged to the order *Erysipelotrichales* (compared to five, three, and one recovered by MetaTOR, bin3C, and HiCBin). Three out of these

eight bins could be annotated at the species level. The other five MAGs could be annotated to four different genus but failed to be annotated at the species level, suggesting the potential expansion of species in the order *Erysipelotrichales*. Further experiments are required to collect more data on their phenotypic and physical properties before these uncultured members can be finally determined.

## Plasmid analyses among high-quality MAGs retrieved by MetaCC binning from the sheep gut sample

Taxonomic statistics of 709 high-quality MAGs retrieved by MetaCC binning from the sheep gut dataset are shown in Supplementary Table 3. Among contigs contained in these high-quality MAGs, 99 contigs were identified as plasmid contigs with high confidence (see Methods). The majority of plasmid contigs were included in MAGs from the orders *Oscillospirales* and *Bacteroidales* (Supplementary Table 4), which were commonly reported in the gut microbiomes[42]. Though there were only 8 out of 709 MAGs (1.1%) from the order *Erysipelotrichales*, 13 out of 99 plasmid contigs (13.1%) could be found within these 8 MAGs. We also observed three plasmid contigs in MAGs from the order *Christensenellales*, members of which are hydrogen-producing fibrolytic and have been reported more predominant in the sheep rumen environment than other rumen environments, such as the mice and rabbits[43].

Plasmids present in multiple copies in genomes are often absent from MAGs retrieved by shotgun-based binning methods since such kind of methods rely on the coverage information to bin contigs[39]. Therefore, we would like to explore whether MetaCC binning could bin multi-copy plasmids. To look for the existence of multi-copy plasmid contigs, we extracted plasmid contigs with coverage > 2 × than the mean average coverage of their respective MAGs, and we observed two plasmids contig_24425 and contig_61128, whose coverages were around 3 × and 5 × than the mean average coverage of their respective MAGs, respectively (Supplementary Table 5). Another plasmid contig_58576 (length: 103,370 bp) had strong BLAST[25] hits with a total of 101,669 bp alignment length (98.4%) to NCBI plasmid reference genome NZ_CP080264.1 (Assigned taxon: *Escherichia coli*). Indeed, MetaCC binning attributed this plasmid contig to BIN 1239, which was annotated as *Escherichia coli* at the species level by GTDB-TK.

## Running time of the overall MetaCC pipeline

On a 2.40 GHz Intel Xeon Processor E5-2665 with 50,000-MB memory allocated, the overall MetaCC pipeline spent 19 min, 56 min, 15 min, and 109 min on the human gut short-read, wastewater short-read, cow rumen long-read, and sheep gut long-read metaHi-C datasets, respectively.

## Discussion

In this work, we have developed MetaCC for scalable and integrative metaHi-C analyses. The MetaCC framework consists of two major modules, the NormCC normalization module and the binning module.

NormCC models both proximity ligation counts across contigs and within contigs using negative binomial and enables correcting all systematic biases. Compared to HiCzin, NormCC showed better performance in terms of the spurious inter-species contact removal and contig clustering on a synthetic yeast dataset and was much faster on real metaHi-C datasets. Moreover, HiCzin suffers substantial performance deterioration when the species-level annotation by TAXAassign is achieved for only a limited fraction of assembled contigs (Supplementary Note 1). This vulnerability is particularly noticeable on long-read metaHi-C datasets (Supplementary Table 1), as further evidenced by HiCzin's failure to converge on the cow rumen long-read metaHi-C dataset (Supplementary Table 2). In contrast, NormCC does not rely on contig annotations as input and performs normalization solely based on fundamental features of assembled contigs, including contig length and the number of restriction sites on contigs. These essential features

can be directly obtained from contigs after assembly regardless of the sequencing technologies employed. This adaptability enables NormCC to be easily applied to both short-read and long-read metaHi-C datasets, demonstrating its versatility in comparison to HiCzin.

MetaCC binning also outperformed all existing Hi-C-based binners consistently on short-read and long-read metaHi-C datasets. Downstream annotation studies and plasmid analyses on real long-read metaHi-C datasets further demonstrated the unique ability of MetaCC on characterizing the structures of microbial samples. Notably, on short-read metaHi-C datasets, HiCBin demonstrated substantial outperformance compared to other competing methods except MetaCC binning, aligning with previous benchmarking studies[30]. MetaCC binning further showed a slight improvement over HiCBin (Fig. 3). Both methods employ Leiden clustering, with the key distinction lying in their respective normalization approaches. MetaCC employs NormCC as its normalization method, whereas HiCBin relies on HiCzin. Consequently, the improved performance of MetaCC binning over HiCBin can be primarily attributed to the superior contig clustering performance facilitated by its normalization method NormCC, as also supported by Fig. 2b. However, in line with HiCzin, HiCBin also exhibits notable degradation in performance when assigning taxonomic labels for contigs at the species level is challenging (Supplementary Note 1), which is particularly evident on long-read metaHi-C datasets (Supplementary Table 1). This limitation of HiCBin adversely affects its performance on long-read metaHi-C datasets, underscoring the notable superiority of MetaCC binning over other Hi-C-based binners, including HiCBin, specifically in the context of long-read metaHi-C datasets.

In the spurious contact removal step, it is important to note that there is no gold standard for determining the threshold value, as the fraction of spurious inter-species contacts among all Hi-C contacts varies due to the quality of metaHi-C experiments. Moreover, there exists a trade-off in selecting this cut-off value. Opting for larger thresholds can eliminate more spurious contacts but may also result in the unintended removal of a higher number of informative intra-species contacts. Therefore, we have taken a conservative approach by selecting a small yet safe threshold (i.e., the default 5-th percentile) to mitigate the loss of important Hi-C information. From our experiments on the synthetic yeast dataset, the default cut-off enabled the removal of 19.3% of spurious inter-species contacts while incorrectly discarding fewer than 0.5% of informative intra-species contacts. Furthermore, we conducted experiments to evaluate the impact of the spurious contact removal step using the default threshold on the downstream binning results of all four real metaHi-C datasets. Our results consistently demonstrated that the inclusion of this step with the default threshold led to improved binning outcomes across all datasets (Supplementary Note 4). In addition to the default conservative thresholds, an outcome-oriented strategy may be an alternative for selecting cut-offs. For example, we can try different thresholds and choose one that yields the best MAG retrieval results in downstream analysis. However, those outcome-oriented strategies always consume much more computing resources and lack generalizability.

In long-read metaHi-C experiments, it is noteworthy that contigs assembled from error-prone long reads are typically polished using accurate short reads obtained from the same sample to improve sequence accuracy[26,27,29]. One important reason for adopting polishing is due to the low alignment quality of accurate Hi-C short reads to contigs assembled from error-prone long reads. Regarding our NormCC normalization method, the polishing step can also help to mitigate the impact of sequencing errors on the identification of restriction sites on contigs due to the improved accuracy at the nucleotide level. However, for the contigs assembled from the accurate HiFi long reads, previous studies have indicated that polishing HiFi assemblies using short reads did not markedly enhance sequence accuracy[28]. Furthermore, we have demonstrated that polishing HiFi

assembles did not improve the Hi-C-based binning results on the sheep gut dataset (Supplementary Note 3). Therefore, considering the high accuracy of HiFi reads, we believe the polishing step may not be necessary in this case.

For the four real metaHi-C datasets, we employed CheckM to evaluate the binning performance. Though CheckM is the main software used to assess the quality of bins retrieved from real metagenomic samples, there is a need for further investigation into how accurately the validation method based on marker genes can reflect the actual completeness and contamination of the recovered MAGs. This is particularly relevant as certain genomic regions may lack marker genes. Moreover, the focus of CheckM on marker sets suitable for evaluating bacterial and archaeal genomes may result in eukaryotic genomes being classified as significantly incomplete[33].

There are several directions that MetaCC can be further extended. For large MAGs with high abundances, it is interesting to combine NormCC-normalized Hi-C contacts with other information sources, such as the assembly graph to scaffold the assembled contigs within the same MAG retrieved by MetaCC binning. Moreover, identifying interactions between mobile genetic elements and hosts using NormCC-normalized Hi-C contacts is of great potential. One major challenge in this topic is to choose a threshold of Hi-C contacts as the true interactions. As a new and the most systematic framework to date, we hope MetaCC enables improved analysis of metaHi-C data with the potential to shed new light on the dark matter of the microbiome.

## Methods

### Real metaHi-C datasets

In this study, we leveraged several publicly available metagenomic Hi-C datasets, consisting of two short-read metaHi-C datasets and two long-read metaHi-C datasets. The specific sizes of raw datasets were shown in Supplementary Table 6.

Two short-read metaHi-C datasets were generated from different microbial ecosystems, including human gut (BioProject: PRJNA413092)[13] and wastewater (BioProject: PRJNA506462)[10]. Each short-read metaHi-C dataset was composed of shotgun libraries and Hi-C libraries derived from the same sample source. The restriction endonucleases Sau3AI and MluCI were utilized to construct all Hi-C sequencing libraries. All shotgun libraries and Hi-C libraries were sequenced by Illumina platforms at 150 bp.

Two long-read metaHi-C datasets were derived from cow rumen samples (BioProject: PRJNA507739)[26] and sheep gut samples (BioProject: PRJNA595610)[28], respectively. The cow rumen long-read metaHi-C dataset consisted of PacBio uncorrected long read libraries and Hi-C libraries. The error-prone PacBio long reads were generated using the PacBio RSII and PacBio Sequel while Hi-C libraries were created by the restriction enzymes Sau3AI and MluCI and subsequently sequenced on an Illumina HiSeq 2000 at 80 bp. The sheep gut long-read metaHi-C dataset contained PacBio circular consensus sequencing (CCS) long read libraries and Hi-C sequencing libraries. PacBio CCS long reads were highly accurate (average Q scores above 20) and hereafter referred to as the HiFi reads. Separate Hi-C libraries from the sheep gut long-read metaHi-C dataset were generated by the restriction endonucleases Sau3AI and MluCI and sequenced at 150 bp for analysis.

### Data processing

In the metagenomic Hi-C experiment, the read cleaning procedure is necessary before the alignment of Hi-C read pairs, since the adaptor sequences, low-quality reads, and PCR duplication can cause significant problems in downstream analyses. Therefore, we applied a standard cleaning procedure to all Hi-C read libraries using bbduk from the BBTools suite (v37.25)[44] (Supplementary Note 5).

For the two short-read metaHi-C datasets, shotgun reads were assembled into contigs by MEGAHIT (v1.2.9)[45] with parameters '-k-min 21 -k-max 141 -k-step 12 -merge-level 20,0.95 -min-contig-len 1000'. The assembled contigs of both PacBio uncorrected long reads and HiFi long reads from the two long-read metaHi-C datasets were provided by the original authors and thus were directly downloaded for analyses. Bickhart et al.[26] assembled PacBio raw reads from the cow rumen long-read metaHi-C dataset by Canu v1.6+101 changes (r8513)[46], and subsequently polished the assembly twice with Illumina data using Pilon[47]. The final assembly was deposited at https://figshare.com/articles/usda_pacbio_second_pilon_indelsonly_fa_gz/8323154. An updated version of the assembly of PacBio HiFi long reads from the long-read sheep gut metaHi-C dataset was provided by authors of the original paper utilizing metaFlye (v2.9)[48] with default parameters and was deposited at https://doi.org/10.5281/zenodo.5228989 under the file 'flye.v29.-sheep_gut.hifi.250g.fasta.gz'. The assembly statistics of contigs from all datasets are shown in Supplementary Table 7.

Finally, we aligned processed paired-end Hi-C reads to assembled contigs by BWA-MEM (v0.7.17)[49]. We switched off the read pairing mode and regarded the alignment with lowest read coordinate as primary alignments with parameter '-5SP' for the BWA-MEM mapping. After the alignment, we successively removed unmapped reads, secondary alignments, supplementary alignments and alignments with low quality (nucleotide match length <30 or mapping score <30). Raw contig-to-contig contacts were aggregated by counting the number of Hi-C read pairs aligned to two contigs separately as across-contig Hi-C contacts, which reflected the proximity extents between contigs. We also defined the number of Hi-C read pairs mapped to the same contig as within-contig Hi-C contacts. Since shorter contigs with fewer Hi-C signals and occurrences of restriction sites tended to have much higher variance, weakening the stability in the downstream analyses[17,18], restrictions on minimum contig length (default, 1000 bp), minimum number of restriction sites (default, one), and minimum Hi-C contacts (default, two across-contig Hi-C contacts and one within-contig Hi-C contact) were imposed to filter problematic contigs. Raw Hi-C contact matrix was then generated from the alignment of Hi-C paired-end reads where the diagonal and non-diagonal entries represented within-contig and across-contig Hi-C contacts, respectively. Notably, because metaHi-C experiments were designed to explore contig-to-contig relationships, across-contig Hi-C contacts were much more important than within-contig Hi-C contacts and unless otherwise specified, Hi-C contacts always referred to across-contig Hi-C contacts in this paper.

### NormCC normalization module in MetaCC

NormCC is a scalable and effective normalization module to eliminate the biases of the number of restriction sites, contig length and coverage on the raw metagenomic Hi-C contacts. Let $H$ denote the raw Hi-C contact matrix. We define the Hi-C signal $M_i$ of contig $i$ as the total number of proximity ligation events between contig $i$ and other contigs, i.e.,

$$M_i = \sum_{k \neq i} H_{ik}. \tag{1}$$

We model the $M_i$ using the negative binomial (NB) distribution, i.e.,

$$M_i \sim \mathrm{NB}(\mu_i, \theta), \tag{2}$$

where $\theta$ is the negative binomial dispersion parameter and the mean $\mu_i$ depends on the three factors of systematic biases for raw metagenomic Hi-C contacts, i.e., the number of restriction sites on contigs, contig length and coverage[14]. Logarithmic link functions in negative binomial regression models[50] are used to model the dependence of

parameter $\mu_i$ on the three factors of biases, i.e.,

$$\log(\mu_i) = \beta_0 + \beta_s \cdot \log(s_i) + \beta_l \cdot \log(l_i) + \beta_c \cdot \log(c_i), \quad (3)$$

where $s_i$, $l_i$, and $c_i$ represent the number of restriction sites, the length, and the coverage of the contig $i$, respectively.

To solve the regression equation (3), we need to obtain the specific values of independent variables, i.e., the three factors of explicit biases for all contigs. Though the number of restriction sites and contig length can be directly obtained, the true contig abundances are always unknown in real datasets. One solution is to estimate the contig coverages by aligning short reads or long reads used in assembly back to contigs. However, the alignment procedure usually consumes a huge amount of computing time and memory resources, especially for long reads[51]. To tackle this problem, we design a statistical model to represent the unknown coverage using known elements. Specifically, let $N_i$ denote the number of proximity ligation events within the contig $i$, i.e.,

$$N_i = H_{ii}. \quad (4)$$

We assume that $N_i$ also follows the negative binomial distribution, i.e.,

$$N_i \sim \text{NB}(\nu_i, \sigma), \quad (5)$$

where $\sigma$ is the negative binomial dispersion parameter and the mean $\nu_i$ is linked to three factors of biases using logarithmic link functions, i.e.,

$$\log(\nu_i) = \gamma_0 + \gamma_s \cdot \log(s_i) + \gamma_l \cdot \log(l_i) + \gamma_c \cdot \log(c_i). \quad (6)$$

Based on formulas (5) and (6), we develop the first negative binomial regression model, denoted by $\text{NBR}_1$, where we consider the factors of systematic biases and the within-contig Hi-C contacts $N_i$ as the predictor variables and the response variable, respectively. The residual of $\text{NBR}_1$ for contig $i$ can be written as

$$N_i/\nu_i. \quad (7)$$

We further assume that no factors other than the number of restriction sites, the length, and the coverage have a major impact on the number of proximity ligation events between fragments within the same contig (i.e., within-contig Hi-C contacts). By taking residuals, the effects of all factors with substantial impacts on the within-contig Hi-C contacts are eliminated. As a result, the residuals described in (7) are primarily composed of non-essential factors, which are assumed to be the same for all contigs, i.e.,

$$\begin{aligned} &N_i/\exp\{\gamma_0 + \gamma_s \cdot \log(s_i) + \gamma_l \cdot \log(l_i) + \gamma_c \cdot \log(c_i)\} \\ &= \frac{N_i}{e^{\gamma_0} s_i^{\gamma_s} l_i^{\gamma_l} c_i^{\gamma_c}} \\ &\doteq C, \end{aligned} \quad (8)$$

where $C$ is a constant. Notably, in addition to factors such as the number of restriction sites, the length, and the coverage of contigs, the extent of spatial proximity across different contigs also plays a major role in determining the number of proximity ligation events between them. Therefore, the assumption mentioned earlier regarding the within-contig Hi-C contacts is not applicable to the across-contig Hi-C contacts.

From formula (8), we can obtain an approximate expression of the contig coverage as

$$c_i \doteq \left( \bar{C} \cdot \frac{N_i}{s_i^{\gamma_s} l_i^{\gamma_l}} \right)^{-\gamma_c}, \quad (9)$$

where $\bar{C} = C^{-1} \cdot e^{-\gamma_0}$.

Therefore, the unknown independent variable $c_i$ can be approximately represented using three observable variables $N_i$, $s_i$, and $l_i$. Though the parameters $\bar{C}$, $\gamma_s$, $\gamma_l$, and $\gamma_c$ are unsolved, we will then show that we don't need to estimate these parameters in our NormCC model.

Let us plug the approximate expression of contig coverage $c_i$ in formula (9) into equation (3), i.e.,

$$\begin{aligned} \log(\mu_i) &= \beta_0 + \beta_s \cdot \log(s_i) + \beta_l \cdot \log(l_i) - \beta_c \gamma_c \cdot \log\left( \bar{C} \cdot \frac{N_i}{s_i^{\gamma_s} l_i^{\gamma_l}} \right) \\ &= (\beta_0 - \beta_c \gamma_c \cdot \log(\bar{C})) + (\beta_s + \beta_c \gamma_c \gamma_s) \cdot \log(s_i) \\ &\quad + (\beta_l + \beta_c \gamma_c \gamma_l) \cdot \log(l_i) - \beta_c \gamma_c \cdot \log(N_i) \\ &= \widetilde{\beta_0} + \widetilde{\beta_s} \cdot \log(s_i) + \widetilde{\beta_l} \cdot \log(l_i) + \widetilde{\beta_N} \cdot \log(N_i), \end{aligned} \quad (10)$$

where

$$\begin{aligned} \widetilde{\beta_0} &= \beta_0 - \beta_c \gamma_c \cdot \log(\bar{C}), \\ \widetilde{\beta_s} &= \beta_s + \beta_c \gamma_c \gamma_s, \\ \widetilde{\beta_l} &= \beta_l + \beta_c \gamma_c \gamma_l, \\ \widetilde{\beta_N} &= -\beta_c \gamma_c. \end{aligned} \quad (11)$$

Based on formulas (2) and (10), we develop the second negative binomial regression model $\text{NBR}_2$. In $\text{NBR}_2$, the Hi-C signal $M_i$ serves as the response variable, while $s_i$, $l_i$, and $N_i$ are considered as predictor variables that contribute to the mean of the distribution $\mu_i$ for a given contig $i$. Since all variables in $\text{NBR}_2$ are observable, we can directly estimate $\widetilde{\beta_0}$, $\widetilde{\beta_s}$, $\widetilde{\beta_l}$, and $\widetilde{\beta_N}$ using the maximum likelihood. Let $\hat{\beta}_0$, $\hat{\beta}_s$, $\hat{\beta}_l$, and $\hat{\beta}_N$ denote the corresponding maximum likelihood estimations. Once the parameters of the model are determined, the estimated mean $\hat{\mu}_i$ can be obtained as

$$\hat{\mu}_i = e^{\hat{\beta}_0} s_i^{\hat{\beta}_s} l_i^{\hat{\beta}_l} N_i^{\hat{\beta}_N}. \quad (12)$$

Notably, from formula (2), $\hat{\mu}_i$ represents the estimated mean of the number of proximity ligation events between contig $i$ and other contigs, while this estimate takes into account only the number of restriction sites, the length, and the coverage of contigs. In other words, $\hat{\mu}_i$ reflects the capability of contig $i$ to produce proximity ligations with other contigs, considering the influence of three bias factors. To address the variations in contig abilities in generating Hi-C interactions due to these bias factors, we normalize the raw Hi-C contacts between contig $i$ and contig $j$ (where $i \neq j$) by dividing them by the square root of $\hat{\mu}_i \cdot \hat{\mu}_j$, i.e.,

$$\frac{H_{ij}}{\sqrt{\hat{\mu}_i \cdot \hat{\mu}_j}} \cdot \hat{C}, \quad (13)$$

where $\hat{C} = \max_k \hat{\mu}_k$ is a rescaling constant. In formula (13), the rescaling constant $\hat{C}$ is used to adjust and scale the values of normalized Hi-C contacts in case they are too small. The square root of $\hat{\mu}_i \cdot \hat{\mu}_j$ can be regarded as a scaled geometric mean of the expected number of proximity ligation events across contigs, predicted only based on three bias factors. Our intuition is that the deviation between the actual across-contig Hi-C contacts and the expected number of proximity ligation events considering only the three bias factors can primarily be attributed to the spatial proximity and thus can reflect the unbiased proximity across contigs.

## Discarding spurious inter-species contacts based on NormCC-normalized Hi-C contacts

Spurious inter-species Hi-C contacts refer to the occurrences of proximity ligation events between contigs from different genomes due to experimental noises and confound the interpretability of the Hi-C data[10]. Based on the expectation that proximity ligations between genomic segments in the same species occur orders of magnitude more frequently than interactions between different species[14], we discard the lowest $p$ percent (default, five) of NormCC-normalized Hi-C contacts as spurious.

## Genome binning in MetaCC

After correcting systematic biases by NormCC and removing spurious Hi-C contacts, the processed Hi-C contact matrix is successively transformed to a weighted graph $\mathcal{G}$ without self-loops where vertices represent all contigs and edge weights are values of NormCC-normalized Hi-C contacts between contigs. Then, we applied the Leiden graph clustering algorithm[32] on the Hi-C contact graph $\mathcal{G}$ to cluster contigs into draft genomic bins. The Leiden algorithm is a modularity-based community detection algorithm and takes greedy strategies to optimize the modularity function. Instead of the classical Newman-Girvan modularity[22] which suffers resolution limits and may fail to identify small bins[23], we leverage a flexible modularity function based on the Reichardt and Bornholdt's Potts model[52] as

$$\sum_{\{i,j|\Delta_{ij}=1\}} \left( e_{ij} - \frac{d_i d_j}{2n} \cdot r \right), \tag{14}$$

where $e_{ij}$ is the edge weight (i.e., NormCC-normalized Hi-C contacts) between contigs $i$ and $j$; $d_i$ and $d_j$ denote the degree of contig $i$ and contig $j$ in the graph $\mathcal{G}$, respectively; $n$ is the total number of edges in the graph; $r$ represents a resolution parameter; $\Delta_{ij}$ is an indicator function and is equal to one if contigs $i$ and $j$ belong to the same community. Notably, the resolution parameter $r$ can be regarded as the relative importance between the configuration null part and links within the communities and controls the number of communities, and the larger $r$ tends to generate more communities[32]. Therefore, determining this hyper-parameter affects the results of contig clustering.

Similar to[53], we detect single-copy marker genes in assembled contigs using FragGeneScan[54] and HMMER (v3.3.2)[55] to estimate the number of genomes in the metagenomic data, denoted by $k$ (Supplementary Note 6). We also set the minimal bin size to the default value of 150 kbp, slightly smaller than the minimum length of known bacterial genomes[56], and consider only contig bins above this size as resolved MAGs. Then, our objective is to select a suitable value of $r$ for which the number of resolved MAGs aligns with the estimated number of genomes in the sample. To achieve this, we sequentially try a list of increasing values for $r$. For each candidate value of the resolution parameter $r$, we record the number of resolved MAGs, denoted as $k_r$. Considering the potential underestimation of the number of genomes, which can occur due to factors such as the possibility of marker genes failing to be detected in certain species, the resolution parameter is determined as the first value for which the number of resolved MAGs surpasses the estimated number of genomes, mathematically, i.e.,

$$\begin{aligned} \min \quad & r \\ s.t. \quad & k_r > k; r \in \{1, 20, 40, 60, 80, \cdots\}. \end{aligned} \tag{15}$$

After selecting the resolution parameter, we can cluster the assembled contigs into MAGs, making up the initial bin set of MetaCC binning.

## Evaluating the quality of recovered MAGs

We applied CheckM (v1.1.3, module: lineage_wf)[33] to evaluate retrieved MAGs. Following the CheckM criteria for completeness and contamination[13], we referred to the resolved MAGs with CheckM completeness greater than or equal to 50% and contamination less than or equal to 10% as high-quality MAGs. We further attributed high-quality draft genomes to three ranks according to the CheckM completeness, i.e., near-complete (completeness ≥ 90% and contamination ≤ 10%), substantially complete (70% ≤ completeness < 90% and contamination ≤ 10%), and moderately complete (50% ≤ completeness < 70% and contamination ≤ 10%).

## Cleaning partially contaminated bins in MetaCC

Apart from high-quality MAGs, there also existed partially contaminated bins with completeness higher than 50% and contamination higher than 10% in the initial bin set of MetaCC binning. Similar to other binners, such as MetaTOR[16] and HiCBin[18], we selected out and cleaned all partially contaminated bins by partitioning contigs within each contaminated bin using the Leiden algorithm. The resolution parameter was kept to be 1 in re-clustering procedures since the number of groups within each partially contaminated bin was expected to be small. As a result, groups of relatively smaller bins, denoted by sub-bins, could be generated and those sub-bins with bin size larger than the minimal requirement (default, 150 kbp) were retained and merged back into the initial bin set to obtain the final bin set of MetaCC binning.

## Assessing the performance of normalization and spurious contact removal on a synthetic yeast metaHi-C dataset

We assessed the normalization performance of NormCC and the following spurious contact removal on an additional synthetic yeast sample (BioProject: PRJNA245328)[11], consisting of 13 yeast species. The synthetic yeast metaHi-C dataset contained shotgun libraries and Hi-C libraries created using restriction enzymes NcoI and HindIII. The raw shotgun and Hi-C libraries contained 85.7 million read pairs at 101 bp and 81 million read pairs at 100 bp, respectively. The read cleaning, contig assembly, and Hi-C read alignment procedures were consistent with those applied to the real short-read metaHi-C datasets. The contig assembly statistics were shown in Supplementary Table 7. Since all species within the synthetic yeast sample were known, the species identity of the assembled contigs could be identified (Supplementary Note 7). Thereafter, the ground truth of intra-species Hi-C contacts (i.e., Hi-C contacts linking contigs from the same species) and spurious inter-species Hi-C (i.e., Hi-C contacts linking contigs from different species) contacts can be generated for benchmarking analyses.

## Estimating the coverages of assembled contigs

Short/long reads used in the assembly were mapped back to assembled contigs to estimate the contigs' abundances. We employed BBMap from the BBTools suite (v37.25)[44] and minimap2 (v2.24)[57] to align short reads and long reads back to contigs, respectively. SAMtools[58] was used to transform the alignment files into bam files, serving as the input for the script 'jgi_summarize_bam_contig_depths' provided by[59] to calculate the contigs' coverages.

## MAG analyses on the human gut short-read metaHi-C dataset

Since many bacteria in the human gut have been identified in previous studies[34,60,61], we evaluated the bins retrieved from the human gut short-read metaHi-C dataset using the repository of known human gut bacteria. Specifically, we downloaded the Unified Human Gastrointestinal Genome (UHGG) database (v1.0)[34], which is one of the largest species-level public gut microbial reference databases. To estimate the number of bins corresponding to known bacteria and the number of bins that might contain chimeric genomes for different binning methods, we utilized Mash (v2.2)[35] with 10,000 sketches per genome to calculate the Mash distance between the UHGG species-level representative and the bins derived from the human gut dataset. Mash distance serves as a reliable proxy for one minus the average

nucleotide identity (ANI)[35], with the Mash species-level threshold of 0.05 equivalent to the widely accepted 95% ANI used to define species boundaries[62]. Therefore, we assigned one bin to one species if the Mash distance between the bin and the representative reference genome of that species was less than 0.05. We also identified a bin as chimeric if it was assigned to multiple species.

### MAG analyses on two long-read metaHi-C datasets

To identify which near-complete bins overlapped each other from MetaCC binning and other Hi-C-based binners, we employed Mash (v2.2)[35] with 10,000 sketches per bin to calculate the Mash distance between near-complete bins from different bin sets. Two bins with mash distance smaller than 0.01 were identified as MAGs from the same genome[31,63]. Moreover, to evaluate ability of different binners to capture the species diversity, we annotated all high-quality bins using GTDB-TK (v2.1.0, Release: R207 v2)[36] with the function 'classify_wf' to obtain the taxonomic information of high-quality MAGs recovered by different binners.

### Plasmid analyses on the sheep gut long-read metaHi-C dataset

A total of 6,320 contigs in 709 high-quality MAGs retrieved by MetaCC binning were first filtered by PPR-Meta (v1.1)[64] with cut-off 0.5 to identify potential plasmid contigs. In this way, we identified 111 (1.7%) potential plasmid contigs. These pre-filtering 111 contigs were further screened using Platon (v1.6)[65] with mode 'Sensitivity' to exclude potential chromosomal contigs. As a result, 99 contigs were finally identified as plasmids with high confidence. We queried these 99 plasmid contigs by BLAST (v2.12.0)[25] with at least 95% identity match of at least 1000 bp to the reference plasmid genomes from NCBI RefSeq database (Release: November, 2022).

### Quality control of all metaHi-C datasets

The quality of Hi-C libraries from different metaHi-C datasets was assessed using qc3C (v0.5)[66] in k-mer mode with default parameters. Results of qc3C for all datasets were shown in Supplementary Data 1.

### Other algorithms used in benchmarking

The normalization method HiCzin (v0.1.0)[14] was run with default parameters. All binners used for comparison, i.e., VAMB (v3.0.3)[31], MetaTOR (v1.1.4)[16], bin3C (v0.1.1)[17], and HiCBin (v1.1.0)[18] were executed with default parameters on all real metaHi-C datasets.

### Reporting summary

Further information on research design is available in the Nature Portfolio Reporting Summary linked to this article.

## Data availability

All the datasets used in this study are publicly available from the NCBI Sequence Read Archive database (http://www.ncbi.nlm.nih.gov/sra). The human gut dataset used in this study is available under accession codes: shotgun library SRR6131123, Hi-C libraries SRR6131122 and SRR6131124. The wastewater dataset is available under accession codes: shotgun library SRR8239393 and Hi-C library SRR8239392. The cow rumen dataset used in this study is available under accession codes: BioProject PRJNA507739. The sheep gut dataset is available under the accession numbers: HiFi reads SRX10647529 and SRX7628648, Hi-C reads SRX10704191, and WGS short reads SRX7649993. The synthetic yeast sample used in this study is available under the accession codes: shotgun library SRR1263009 and Hi-C library SRR1262938. The final assembly from the cow rumen dataset is available at https://figshare.com/articles/usda_pacbio_second_pilon_indelsonly_fa_gz/8323154. The assembly of PacBio HiFi long reads from the sheep gut dataset is available at https://doi.org/10.5281/zenodo.5228989 under the file 'flye.v29.sheep_gut.hifi.250g.fasta.gz'. The curated nucleotide reference database of TAXAassign is available at http://userweb.eng.gla.ac.uk/

umer.ijaz/bioinformatics/db.sqlite.gz. The GTDB-TK reference database is available at https://data.gtdb.ecogenomic.org/releases/release207/207.0/auxillary_files/gtdbtk_r207_v2_data.tar.gz. The UHGG catalogs are available from the MGnify FTP site http://ftp.ebi.ac.uk/pub/databases/metagenomics/mgnify_genomes/human-gut/v1.0/uhgg_catalogue. The NCBI RefSeq database is available at https://ftp.ncbi.nlm.nih.gov/refseq/release. The complete sequence of NCBI plasmid reference genome NZ_CP080264.1 is available at https://www.ncbi.nlm.nih.gov/nuccore/NZ_CP080264.1. The MAGs generated by MetaCC binning from real metagenomes in the benchmarking can be obtained from Zenodo: https://doi.org/10.5281/zenodo.8057996[67]. The remaining data are available within the Article, Supplementary Information, or Source data. There is no restriction on data availability. Source data are provided with this paper.

## Code availability

The MetaCC software is freely available at https://github.com/dyxstat/MetaCC under the GNU General Public License version v3. The MetaCC code used in this work is also archived on Zenodo under https://doi.org/10.5281/zenodo.8054563[68]. Scripts used in this study to process the intermediate data and plot figures are available at https://github.com/dyxstat/Reproduce_MetaCC.

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

## Acknowledgements

The research is partially funded by NIH grant R01GM131407 and NSF grant EF-2125142.

## Author contributions

Y.D. and F.S. conceived the ideas and designed the study. Y.D. implemented the methods, carried out the computational analyses, and drafted the manuscript. F.S. and Y.D. modified and finalized the paper.

## Competing interests

The authors declare no competing interests.
