## [Peer Review File · Nature Communications]

MetaCC allows scalable and integrative analyses of both long-read and short-read metagenomic Hi-C dataREVIEWER COMMENTS

Reviewer #1 (Remarks to the Author):

This paper developed a new framework, “MetaCC,” to improve contig binning for short-read and long-read metaHi-C data. A major contribution is a new normalization method that can effectively eliminate systematic biases in the raw Hi-C contact matrix. The authors demonstrated that this normalisation module NormCC can achieve better performance in removing spurious contacts caused by experimental noise. Besides NormCC, another contribution is MetaCC’s application to long reads. The paper is organised and well-written. The experimental results on the yeast data and two real datasets demonstrate that MetaCC can generate more complete bins, which can benefit both composition and function analysis for metagenomic data. Overall, I enjoyed reading this paper. Nevertheless, I do have some specific questions or suggestions that need to be clarified/elaborated.

Questions about TGS

I agree with the authors on the importance of handling metaHi-C data generated using TGS. But I did not find discussions about the adaptations of NormCC to long read/short read assemblies. Contigs assembled from NGS and TGS may exhibit some differences in length, coverage, error rate, GC bias etc. Will the differences affect NormCC? Will NormCC incorporate these differences? For example, given a set of longer contigs with slightly higher error rates, will the steps such as finding the restriction sites/annotation be affected? If the data is not generated by HiFi but some other protocols with slightly higher error rate, will NormCC be affected?

Introduction

Line 087. To make MetaCC more self-contained, it will be helpful to briefly explain “annotation” needed by HiCBin. Annotation is used multiple times in the main draft but I could not find the specific definition/explanation of this step. Does it refer to protein/gene annotation?

This question also applies to page 8, line 335.

Overall, I think the authors can explain why NormCC is better than HiCzin from the design of the normalisation methods. It will help readers appreciate NormCC more and link the performance improvement to the model.

Method:

1. To make the paper reach more general audience, it would be helpful if the authors could provide more explanations or justifications for some equations/symbols.

a) The definition of theta and delta in formula (2) and (5).

b) The idea of using another NB with known variables to present the hard-to-estimate variable (c_i) is clear. However, the process of how to present c_i is too concise to follow: (1) "once we remove the effects of ..., the residues are approximately the same for all contigs" at rows 592 and after, how do you derive the residues and why the residues are the same for all contigs? (2) Can you explain this more clearly in the sentence "Notably, the above assumption ... becomes dominant factor" in row 602?

c) The variable "u" is associated with three factors of systemic biases. There is a jump from "u" to normalized Hi-C contact by the formula (11). Why " H_{ij} " divided by the square root of ($u_i * u_j$) and multiplied with the maximum value of "u" can remove the biases? I would suggest authors explain formula (11) more specifically.

2. The authors proposed a method to determine the parameter "r" automatically. With the increasing value of "r," the smallest value with the number of resolved MAGs exceeding the initial bin number is the adopted value. If the number of resolved MAGs increases with the increase of "r"? If yes, can you output the value of "r" when the number of MAGs does not increase significantly?

Result:

3. As most bacteria in human gut have been identified in related studies, the authors can get more accurate evaluation of the bins using the catalog of human gut bacteria. For example, is it possible to do a more accurate evaluation of these bins in Fig. 3 (left panel) using the known bacteria in human gut? How many of the bins by each tool are correct (corresponding to known bacteria) and how many might contain chimeric genomes?

4. I also suggest that the authors add discussions about the limitations of CheckM in Discussion. As CheckM is the main software used to evaluate different binning tools on the real datasets, it will be helpful to see whether it may produce some errors.

5. Is it possible to include other Hi-C based binning tools in Fig. 2 (a)?

Minor:

1. In the sentence at Row 560, "...the dependence of parameter v_i ", is that " v_i " should be " u_i "?

2. Section 2.2.2, Line 196/197, this sentence can be polished. "incorrectly removing fewer..." reads a little funny.

Reviewer #2 (Remarks to the Author):

Comment

In this nicely written manuscript, the authors proposed a new metagenomic binning pipeline, MetaCC, for metaHi-C. Using synthetic yeast data and real datasets of different properties, the authors demonstrated a significant improvement over other state-of-the-art methods in both accuracy and computational efficiency. The improvement over long-read metagenomic Hi-C data is even more remarkable. Overall, I believe that this a nice and solid work. The topic is highly significant since many investigators are using the metaHi-C technology to study the microbiome now. The novel normalization method, simple yet powerful, makes the method highly scalable. The validation is also thorough and rigorous. It is surely a nice contribution to the field. I have only several minor comments the authors may consider.

1. The authors may provide more intuition and rationale as why the performance on long-read data is much better than competing methods.
2. The authors briefly discussed the method for discarding spurious inter-species contacts. It uses five percent threshold to declare those contacts as “Spurious”. Any rationale? Any impact on the downstream binning?
3. Can the full computational time (not just the time for normalization) for the real datasets be provided?

Response to the first reviewer's comments

This paper developed a new framework, "MetaCC," to improve contig binning for short-read and long-read metaHi-C data. A major contribution is a new normalization method that can effectively eliminate systematic biases in the raw Hi-C contact matrix. The authors demonstrated that this normalisation module NormCC can achieve better performance in removing spurious contacts caused by experimental noise. Besides NormCC, another contribution is MetaCC's application to long reads. The paper is organised and well-written. The experimental results on the yeast data and two real datasets demonstrate that MetaCC can generate more complete bins, which can benefit both composition and function analysis for metagenomic data. Overall, I enjoyed reading this paper. Nevertheless, I do have some specific questions or suggestions that need to be clarified/elaborated.

Thank you for your time and effort in reviewing our manuscript and for your positive feedback. We are delighted to hear that you found our MetaCC framework to be valuable contributions and we sincerely appreciate your important and constructive comments! Your suggestions have significantly helped us improve our manuscript. In the revision, we have carefully followed your insightful comments and made detailed efforts to address each point, providing further clarification and elaboration as outlined below.

1. I agree with the authors on the importance of handling metaHi-C data generated using TGS. But I did not find discussions about the adaptations of NormCC to long read/short read assemblies. Contigs assembled from NGS and TGS may exhibit some differences in length, coverage, error rate, GC bias etc. Will the differences affect NormCC? Will NormCC incorporate these differences? For example, given a set of longer contigs with slightly higher error rates, will the steps such as finding the restriction sites/annotation be affected? If the data is not generated by HiFi but some other protocols with slightly higher error rate, will NormCC be affected?

Response: This is an excellent point! We fully acknowledge the reviewer's comments regarding the inherent differences between contigs assembled from NGS and TGS metaHi-C datasets, and the differences indeed affect contig annotations (i.e., assign contigs to various taxonomic levels), resulting in substantial performance deterioration of the state-of-the-art metaHi-C-based methods. Specifically, we observe that the performances of HiCBin, which demonstrated the superior binning performance on short-read metaHi-C datasets according to most recent benchmarking studies (Jia et al., 2023), are markedly deteriorated on long-read metaHi-C datasets (see Subsection 2.4 and Fig. 4). One essential factor contributing to this decline is the large degradation of HiCBin as well as its adopted normalization method HiCzin when only a small fraction of assembled contigs can be successfully labeled at the species level by TAXAassign (see Supplementary Note 1). Additionally, the taxonomic labeling of contigs assembled from long reads poses a challenge for TAXAassign (see Supplementary Table 2), consequently limiting the effectiveness of HiCzin and HiCBin on long-read metaHi-C datasets.

Given that addressing this challenge serves as one of the primary motivations behind the development of new state-of-the-art computational methods that can be adaptable to both short-read and long-read metaHi-C datasets, we have incorporated the above discussions into the **Introduction** section to provide a clearer explanation of our underlying motivation (see Page 3 Line 95).

In comparison to HiCzin, NormCC does not rely on contig annotations as input and performs normalization solely based on fundamental features of assembled contigs, including contig length and the number of restriction sites on contigs. These essential features can be directly obtained from contigs after assembly regardless of the sequencing technologies employed. Specifically, the contig length is one of the most essential features of assembled contigs and can be directly observed regardless of the sequencing technologies utilized. Regarding the number of restriction sites, we acknowledge the reviewer's concern that error-prone long reads may impact the identification of restriction sites on contigs. However, based on the currently available metaHi-C datasets utilizing error-prone long reads, it is noteworthy that contigs assembled from error-prone long reads are typically polished using accurate short reads obtained from the same sample to improve sequence accuracy in long-read metaHi-C studies (Bickhart et al., 2019; Cuscó et al., 2022; Gounot et al., 2022). One important reason for adopting polishing is due to the low alignment quality of accurate Hi-C short reads to contigs assembled from error-prone long reads. Consequently, we believe that after polishing, the error rate associated with error-prone long reads will not significantly affect the identification of restriction sites on contigs due to the improved accuracy at the nucleotide level. In conclusion, the minimal impact of sequencing technologies on the input information required by NormCC for normalization in metaHi-C experiments underscores its adaptability and versatility.

We sincerely appreciate the reviewer for this crucial aspect, as it further highlights the advantage of NormCC as a method that can effectively handle challenges introduced by different sequencing platforms compared to HiCzin. Therefore, we have added the above discussions about the adaptability and versatility of NormCC to the revised manuscript in the **Discussion** section as follows (see Page 20 Line 897):

'Moreover, HiCzin suffers substantial performance deterioration when the species-level annotation by TAXAassign is achieved for only a limited fraction of assembled contigs (Supplementary Note 1). This vulnerability is particularly noticeable on long-read metaHi-C datasets (Supplementary Table 2), as further evidenced by HiCzin's failure to converge on the cow rumen long-read metaHi-C dataset (Supplementary Table 1). In contrast, NormCC does not rely on contig annotations as input and performs normalization solely based on fundamental features of assembled contigs, including contig length and the number of restriction sites on contigs. These essential features can be directly obtained from contigs after assembly regardless of the sequencing technologies employed. This adaptability enables NormCC to be easily applied to both short-read and long-read metaHi-C datasets, demonstrating its versatility in comparison to HiCzin.'

We have also incorporated the discussion about the impact of error-prone long reads on metaHi-C experiments and NormCC, as well as the discussion about the contig polishing in long-read metaHi-C studies in the **Discussion** section as follows (Page 21 Line 954):

'In long-read metaHi-C experiments, it is noteworthy that contigs assembled from error-prone long reads are typically polished using accurate short reads obtained from the same sample to improve sequence accuracy [25, 26, 28]. One important reason for adopting polishing is due to the low alignment quality of accurate Hi-C short reads to contigs assembled from error-prone long reads. Regarding our NormCC normalization method, the polishing step can also help to mitigate the impact of sequencing errors on the identification of restriction sites on contigs due to the improved accuracy at the nucleotide level. However, for the contigs assembled from the accurate HiFi long reads, previous studies have indicated that polishing HiFi assemblies using short reads did not markedly enhance sequence accuracy [27]. Furthermore, we have demonstrated that polishing HiFi assemblies did not improve the Hi-C-based binning results on the sheep gut dataset (Supplementary Note 3). Therefore, considering the high accuracy of HiFi reads, we believe the polishing step may not be necessary in this case.'

2. Line 087. To make MetaCC more self-contained, it will be helpful to briefly explain "annotation" needed by HiCBin. Annotation is used multiple times in the main draft but I could not find the specific definition/explanation of this step. Does it refer to protein/gene annotation? This question also applies to page 8, line 335.

Response: Thanks for the useful comment! We have added the definition of the term 'annotation' used in the revised manuscript and briefly explained the annotation step needed by HiCBin as follows (Page 2 Line 85):

'... and generating contig annotations, which refers to assigning nucleotide sequences to various taxonomic levels. Specifically, HiCzin and HiCBin utilize TAXAassign [23] to label contigs at the species level by running BLAST [24] against a curated nucleotide reference database.'

To avoid repetition, we have removed the sentence referring to 'annotated contigs' from the original Page 8 Line 335 (see Page 9 Line 373 in the revised manuscript), as we have provided more detailed discussions in **Discussion** section (see Page 20 Line 915).

3. Overall, I think the authors can explain why NormCC is better than HiCzin from the design of the normalisation methods. It will help readers appreciate NormCC more and link the performance improvement to the model.

Response: Thanks for this insightful point, which greatly enhances the elaboration of our NormCC method! Following your suggestion, we have included several sentences in the **Introduction** section to explain why NormCC is better than HiCzin from the aspect of model design as follows (Page 3 Line 113):

'In comparison to HiCzin, which relies on estimated contig abundances as input, NormCC employs a negative binomial regression model to represent contig abundances based on easily obtainable features including the number of restriction sites on contigs, contig length, and the number of proximity ligation events within contigs. Consequently, NormCC does not require the estimation of contig abundances. Additionally, HiCzin models the Hi-C contacts between contigs of the same species, necessitating contig annotations. Conversely, NormCC models the total number of proximity ligation events for each contig using a second negative binomial regression, eliminating the need for contig annotation.'

4. To make the paper reach more general audience, it would be helpful if the authors could provide more explanations or justifications for some equations/symbols.

a) The definition of theta and delta in formula (2) and (5).

Response: Thanks for the helpful comments! We have added the definition of θ and σ , which is the negative binomial dispersion parameter, in formulas (2) and (5) (see Page 13 Line 592 and Page 14 Line 618).

b) The idea of using another NB with known variables to present the hard-to-estimate variable (c_i) is clear. However, the process of how to present c_i is too concise to follow: (1) "once we remove the effects of ..., the residues are approximately the same for all contigs" at rows 592 and after, how do you derive the residues and why the residues are the same for all contigs? (2) Can you explain this more clearly in the sentence "Notably, the above assumption ... becomes dominant factor" in row 602?

Response: We appreciate these valuable suggestions. Regarding suggestion (1), we have incorporated detailed explanations on how to represent the contig abundances using known variable as follows (Page 14 Line 623):

'Based on formulas (5) and (6), we develop the first negative binomial regression model, denoted by NBR_1 , where we consider the factors of systematic biases and the within-contig Hi-C contacts N_i as the predictor variables and the response variable, respectively. The residual of NBR_1 for contig i can be written as

$$N_i/v_i. \tag{7}$$

We further assume that no factors other than the number of restriction sites, the length, and the coverage have a major impact on the number of proximity ligation events between fragments within the same contig (i.e., within-contig Hi-C contacts). By taking residuals, the effects of all factors with substantial impacts on the within-contig Hi-C contacts are eliminated. As a result,

the residuals described in (7) are primarily composed of non-essential factors, which are assumed to be the same for all contigs, i.e.,

$$\begin{aligned}
 & N_i / \exp\{\gamma_0 + \gamma_s \cdot \log(s_i) + \gamma_l \cdot \log(l_i) + \gamma_c \cdot \log(c_i)\} \\
 &= \frac{N_i}{e^{\gamma_0} s_i^{\gamma_s} l_i^{\gamma_l} c_i^{\gamma_c}} \\
 &\doteq C,
 \end{aligned} \tag{8}$$

where C is a constant.'

Regarding suggestion (2), we have rewritten the original sentence and added more details to provide clearer explanations as follow (Page 15 Line 645):

'Notably, in addition to factors such as the number of restriction sites, the length, and the coverage of contigs, the extent of spatial proximity across different contigs also plays a major role in determining the number of proximity ligation events between them. Therefore, the assumption mentioned earlier regarding the within-contig Hi-C contacts is not applicable to the across-contig Hi-C contacts.'

c) The variable “u” is associated with three factors of systemic biases. There is a jump from “u” to normalized Hi-C contact by the formula (11). Why “H_ij” divided by the square root of (u_i*u_j) and multiplied with the maximum value of “u” can remove the biases? I would suggest authors explain formula (11) more specifically.

Response: Thanks for this helpful comment, and we agree that providing more explanations of original formula (11) can greatly enhance readers’ understanding of our method. Following your suggestion, we have included our motivations and intuitions behind the original formula (11) and provided more detailed explanations on how to derive it as follows (Page 15 Line 680):

'Based on formulas (2) and (10), we develop the second negative binomial regression model NBR₂. In NBR₂, the Hi-C signal M_i serves as the response variable, while s_i, l_i, and N_i are considered as predictor variables that contribute to the mean of the distribution μ_i for a given contig i. Since all variables in NBR₂ are observable, we can directly estimate β̂₀, β̂_s, β̂_l, and β̂_N using the maximum likelihood. Let β̂₀, β̂_s, β̂_l, and β̂_N denote the corresponding maximum likelihood estimations. Once the parameters of the model are determined, the estimated mean μ̂_i can be obtained as

$$\hat{\mu}_i = e^{\hat{\beta}_0} s_i^{\hat{\beta}_s} l_i^{\hat{\beta}_l} N_i^{\hat{\beta}_N}. \tag{12}$$

Notably, from formula (2), μ̂_i represents the estimated mean of the number of proximity ligation events between contig i and other contigs, while this estimate takes into account only the number of restriction sites, the length, and the coverage of contigs. In other words, μ̂_i reflects the capability of contig i to produce proximity ligations with other contigs, considering the influence of three bias factors. To address the variations in contig abilities in generating Hi-C

interactions due to these bias factors, we normalize the raw Hi-C contacts between contig i and contig j (where $i \neq j$) by dividing them by the square root of $\hat{\mu}_i \cdot \hat{\mu}_j$, i.e.,

$$\frac{H_{ij}}{\sqrt{\hat{\mu}_i \cdot \hat{\mu}_j}} \cdot \hat{C}, \quad (13)$$

where $\hat{C} = \max_k \hat{\mu}_k$ is a rescaling constant. In formula 13, the rescaling constant \hat{C} is used to adjust and scale the values of normalized Hi-C contacts in case they are too small. The square root of $\hat{\mu}_i \cdot \hat{\mu}_j$ can be regarded as a scaled geometric mean of the expected number of proximity ligation events across contigs, predicted only based on three bias factors. Our intuition is that the deviation between the actual across-contig Hi-C contacts and the expected number of proximity ligation events considering only the three bias factors can primarily be attributed to the spatial proximity and thus can reflect the unbiased proximity across contigs.'

5. The authors proposed a method to determine the parameter “r” automatically. With the increasing value of “r,” the smallest value with the number of resolved MAGs exceeding the initial bin number is the adopted value. If the number of resolved MAGs increases with the increase of “r”? If yes, can you output the value of “r” when the number of MAGs does not increase significantly?

Response: Thank you for providing your valuable comment! Though the larger resolution parameter ‘r’ tends to generate more communities (Traag et al., 2019) (Page 17 Line 747), we observed that the number of resolved MAGs does not consistently increase with the increase of ‘r’ from our experiments. For example, on the cow rumen dataset, we observed that the number of MAGs decreased from 981 to 914 when ‘r’ increased from 200 to 300. Given the lack of theoretical assurance regarding the convergence of the number of resolved MAGs concerning ‘r’, it becomes challenging for us to provide a specific value of ‘r’ when the number of MAGs does not show a significant increase.

Nevertheless, we acknowledge the importance of this comment as it has helped us realize that the term ‘initial bin number’ in the original manuscript might be confusing. In fact, the original term ‘initial bin number’ refers to an estimation of the number of genomes present in the metagenomic data. It is also crucial to emphasize that the number of MAGs carries important biological meaning. Specifically, in the context of contig binning problems, the number of resolved MAGs should be determined based on the actual number of genomes existing in the microbial sample. Since the true number of genomes in microbial samples is often unknown in real metagenomic data, previous methods (Wu et al., 2016; Lu et al., 2017; Wang et al., 2019; Wang et al., 2023) have employed the strategy proposed by MaxBin (Wu et al., 2014) of utilizing single-copy marker genes to estimate this important number. In MetaCC binning, we also adopt the same strategy to estimate the number of genomes in the metagenomic data. Subsequently, our aim and objective is to select a suitable value of “r” for which the number of resolved MAGs aligns with the estimated number of genomes.

We are sorry that we didn't clarify our motivation to determine the resolution parameter clearly in our original manuscript, leading to the confusion. We have revised the corresponding paragraph as follows (Page 17 Line 752):

'Similar to [52], we detect single-copy marker genes in assembled contigs using FragGeneScan [53] and HMMER (v3.3.2) [54] to estimate the number of genomes in the metagenomic data, denoted by k (Supplementary Note 5). We also set the minimal bin size to the default value of 150 kbp, slightly smaller than the minimum length of known bacterial genomes [55], and consider only contig bins above this size as resolved MAGs. Then, our objective is to select a suitable value of r for which the number of resolved MAGs aligns with the estimated number of genomes in the sample. To achieve this, we sequentially try a list of increasing values for r . For each candidate value of the resolution parameter r , we record the number of resolved MAGs, denoted as k_r . Considering the potential underestimation of the number of genomes, which can occur due to factors such as the possibility of marker genes failing to be detected in certain species, the resolution parameter is determined as the first value for which the number of resolved MAGs surpasses the estimated number of genomes.'

We have also briefly described how to estimate the number of genomes existing in the metagenomic data in the Supplementary Note 5 as follows:

'Supplementary Note 5: Estimating the number of genomes existing in the metagenomic data using single-copy marker genes

Following the strategy in [3], we utilized single-copy marker genes to estimate the number of genomes in the microbial sample. To predict genes from the contigs, FragGeneScan [4] was employed, and the predicted genes were scanned using HMMER3 (v3.3.2) [5] with parameter '-cut_tc' to identify 107 single-copy marker genes that are conserved in 95% of sequenced bacteria [6]. After filtering out genes that do not meet the coverage threshold (set at 40%), we determined the number of genomes present in the metagenomic data k as the median number of contigs containing each of the marker genes. This step accounted for the possibility of marker genes being fragmented into multiple pieces, which could affect the estimation of the number of genomes.'

6. As most bacteria in human gut have been identified in related studies, the authors can get more accurate evaluation of the bins using the catalog of human gut bacteria. For example, is it possible to do a more accurate evaluation of these bins in Fig. 3 (left panel) using the known bacteria in human gut? How many of the bins by each tool are correct (corresponding to known bacteria) and how many might contain chimeric genomes?

Response: This is an excellent point! Following the insightful suggestion of the reviewer, we downloaded the Unified Human Gastrointestinal Genome (UHGG) database (Almeida et al., 2021), which is one of the most widely used and largest reference databases for the human gut

microbiome. Notably, though many bacteria in the human gut have been identified in previous studies, the reference genomes for the majority of these bacteria are still putative draft metagenome-assembled genomes (i.e., MAGs) derived from metagenomic contig binning, as we have done in our manuscript, and consist of fragmented contigs or scaffolds. For example, the UHGG reference database contains representative reference genomes of 4,644 species. However, 3,750 (81%) of the UHGG species lack cultured representatives and their reference genomes are fragmented MAGs. Even among the 894 species in UHGG that have cultured representatives, their representative reference genomes are incomplete, with 731 out of 894 representatives comprising more than ten contigs/scaffolds.

Considering the incomplete, fragmented, and putative property of the reference genomes for most bacteria from the human gut environment, conducting alignment-based analyses of contig bins derived from different binning tools is challenging. Instead, similar to previous studies (e.g., Groussin et al., 2021; Tomofuji et al., 2022), we can evaluate the number of bins that correspond to known bacteria and the number of bins that might contain chimeric genomes using Mash (Ondov et al., 2016), a general-purpose toolkit that utilizes the MinHash technique to estimate genomic distance. Mash distance serves as a reliable proxy for one minus the average nucleotide identity (ANI), with the Mash species-level threshold of 0.05 equivalent to the widely accepted 95% ANI used to define species boundaries.

Specifically, we employed Mash (v2.2) with 10,000 sketches per genome to calculate the Mash distance between the representative reference genomes in UHGG and the bins derived from different binning tools. We assigned one bin to one species if the Mash distance between the bin and the representative genome of that species was less than 0.05. We also identified a bin as chimeric if it was assigned to multiple species. As a result, we observed that MetaCC binning was able to recover the largest number of known bacteria from the human gut environment based on UHGG reference database. More specifically, VAMB, MetaTOR, bin3C, HiCBin, and MetaCC binning retrieved 83, 107, 89, 118, and 128 bins, respectively, which were assigned to only one known species. Furthermore, only one bin was detected as chimeric for MetaCC binning, while 4, 6, 2, and 11 chimeric bins were identified for VAMB, MetaTOR, bin3C, and HiCBin, respectively. These results further demonstrated the superior performance of MetaCC binning in comparison to other binning tools.

We sincerely thank the reviewer for this important comment for strengthening our results. We have added the methods discussed above in our revised manuscript as follows (Page 19 Line 829):

'3.10 MAG analyses on the human gut short-read metaHi-C dataset

Since many bacteria in the human gut have been identified in previous studies [33, 59, 60], we evaluated the bins retrieved from the human gut short-read metaHi-C dataset using the repository of known human gut bacteria. Specifically, we downloaded the Unified Human Gastrointestinal Genome (UHGG) database [33], which is one of the largest species-level public gut microbial reference databases. To estimate the number of bins corresponding to known

bacteria and the number of bins that might contain chimeric genomes for different binning methods, we utilized Mash (v2.2) [34] with 10,000 sketches per genome to calculate the Mash distance between the UHGG species-level representative and the bins derived from the human gut dataset. Mash distance serves as a reliable proxy for one minus the average nucleotide identity (ANI) [34], with the Mash species-level threshold of 0.05 equivalent to the widely accepted 95% ANI used to define species boundaries [61]. Therefore, we assigned one bin to one species if the Mash distance between the bin and the representative reference genome of that species was less than 0.05. We also identified a bin as chimeric if it was assigned to multiple species.'

The results have been shown in the revised manuscript as follows (Page 7 Line 312):

'Additionally, for the human gut short-read metaHi-C dataset, we assessed the number of bins corresponding to known bacteria and the number of bins that might contain chimeric genomes for different binning methods using UHGG gut microbial reference database [33] (see Subsection 3.10). MetaCC binning recovered the largest number of known bacteria from the human gut environment based on the UHGG database. Specifically, VAMB, MetaTOR, bin3C, HiCBin, and MetaCC binning retrieved 83, 107, 89, 118, and 128 bins, respectively, which were assigned to only one known species. Furthermore, only one bin was detected as chimeric for MetaCC binning, while 4, 6, 2, and 11 chimeric bins were identified for VAMB, MetaTOR, bin3C, and HiCBin, respectively.'

7. I also suggest that the authors add discussions about the limitations of CheckM in Discussion. As CheckM is the main software used to evaluate different binning tools on the real datasets, it will be helpful to see whether it may produce some errors.

Response: Thanks for this helpful comment, and we have added the discussions about the limitations of CheckM in Discussion section as follows (Page 22 Line 970):

'For the four real metaHi-C datasets, we employed CheckM to evaluate the binning performance. Though CheckM is the main software used to assess the quality of bins retrieved from real metagenomic samples, there is a need for further investigation into how accurately the validation method based on marker genes can reflect the actual completeness and contamination of the recovered MAGs. This is particularly relevant as certain genomic regions may lack marker genes. Moreover, the focus of CheckM on marker sets suitable for evaluating bacterial and archaeal genomes may result in eukaryotic genomes being classified as significantly incomplete [32].'

8. Is it possible to include other Hi-C based binning tools in Fig. 2 (a)?

Response: Thank you for providing this valuable comment! The reason we did not include other metaHi-C normalization methods is that the comparison of additional methods has already been addressed in our previous paper (Du et al., 2022), which used the same yeast metaHi-C dataset and employed the same comparison criteria. In that paper, we demonstrated that HiCzin, the normalization method employed in HiCBin, markedly outperformed other metaHi-C-based normalization methods, including those utilized in bin3C and MetaTOR, in terms of spurious contact removal and contig clustering, and HiCzin is the only method that can eliminate all systematic biases. To avoid redundancy and maintain focus on the comparison between HiCzin and our newly designed NormCC, we did not present the results of other metaHi-C normalization methods. However, we acknowledge that this is a valid point, as readers may wonder why we did not include other methods. Therefore, we have added several sentences to explain this in our revised manuscript as follows (Page 5 Line 193):

'We have also demonstrated that HiCzin, the normalization method employed in HiCBin, outperformed other metaHi-C-based normalization methods, including those utilized in bin3C and MetaTOR, in terms of spurious contact detection and contig clustering using the synthetic yeast metaHi-C dataset [13, 17]. Notably, HiCzin incorporates contig annotations at the species level, obtained through TAXAassign [23], to select intra-species Hi-C contacts utilized in fitting its normalization model. In line with the previous analyses, we validated the performance of NormCC normalization on this synthetic sample and compared it to HiCzin using the same benchmarking criteria.'

9. In the sentence at Row 560, "...the dependence of parameter v_i ", is that " v_i " should be " u_i "?

Response: Thanks, revised (Page 13 Line 596)!

10. Section 2.2.2, Line 196/197, this sentence can be polished. "incorrectly removing fewer..." reads a little funny.

Response: Thanks, we have polished this sentence as follows (Page 5 Line 225):

'Therefore, the improved capacity for removing spurious contacts from one single Hi-C contact matrix can be assessed by effectively eliminating a greater number of spurious contacts while minimizing the unintended removal of informative intra-species contacts.'

Response to the second reviewer's comments

In this nicely written manuscript, the authors proposed a new metagenomic binning pipeline, MetaCC, for metaHi-C. Using synthetic yeast data and real datasets of different properties, the authors demonstrated a significant improvement over other state-of-the-art methods in both accuracy and computational efficiency. The improvement over long-read metagenomic Hi-C data is even more remarkable. Overall, I believe that this is a nice and solid work. The topic is highly significant since many investigators are using the metaHi-C technology to study the microbiome now. The novel normalization method, simple yet powerful, makes the method highly scalable. The validation is also thorough and rigorous. It is surely a nice contribution to the field. I have only several minor comments the authors may consider.

Thank you for your positive feedback and thoughtful review of our manuscript. We sincerely appreciate your recognition of our MetaCC framework and your insightful comments! Your suggestions have greatly helped us improve our manuscript. We have carefully addressed each point in detail as outlined below in response to your helpful suggestions.

1. The authors may provide more intuition and rationale as why the performance on long-read data is much better than competing methods.

Response: Thanks for providing this helpful comment! We have expanded on the intuition and rationale to elucidate the superior performance of MetaCC binning on long-read metaHi-C datasets compared to other Hi-C-based binning methods, particularly HiCBin, which demonstrated the superior binning performance on short-read metaHi-C datasets according to recent benchmarking studies (Jia et al., 2023), in the **Discussion** section as follows (Page 20 Line 914):

'Notably, on short-read metaHi-C datasets, HiCBin demonstrated substantial outperformance compared to other competing methods except MetaCC binning, aligning with previous benchmarking studies [29]. MetaCC binning further showed a slight improvement over HiCBin (Fig. 3). Both methods employ Leiden clustering, with the key distinction lying in their respective normalization approaches. MetaCC employs NormCC as its normalization method, whereas HiCBin relies on HiCzin. Consequently, the improved performance of MetaCC binning over HiCBin can be primarily attributed to the superior contig clustering performance facilitated by its normalization method NormCC, as also supported by Fig. 2b. However, in line with HiCzin, HiCBin also exhibits notable degradation in performance when assigning taxonomic labels for contigs at the species level is challenging (Supplementary Note 1), which is particularly evident on long-read metaHi-C datasets (Supplementary Table 2). This limitation of HiCBin adversely affects its performance on long-read metaHi-C datasets, underscoring the notable superiority of MetaCC binning over other Hi-C-based bidders, including HiCBin, specifically in the context of long-read metaHi-C datasets.'

2. The authors briefly discussed the method for discarding spurious inter-species contacts. It uses five percent threshold to declare those contacts as “Spurious”. Any rationale? Any impact on the downstream binning?

Response: This is an excellent point! It is important to acknowledge that there is no gold standard for determining the threshold value, as the fraction of spurious inter-species contacts among all Hi-C contacts varied due to the quality of metaHi-C experiments. Furthermore, there exists a trade-off in selecting this cut-off value. Opting for larger thresholds can eliminate more spurious contacts but may also result in the unintended removal of a higher number of informative intra-species contacts. Therefore, we have taken a conservative approach by selecting a small yet safe threshold (i.e., the default 5-th percentile) to mitigate the loss of important Hi-C information. From our experiments on the synthetic yeast dataset, the default cut-off enables the removal of 19.3% of spurious inter-species contacts while incorrectly discarding less than 0.5% of informative intra-species contacts. To assess the impact of the spurious contact detection step using our default threshold on the subsequent binning process, we executed the MetaCC pipeline without the spurious contact removal step. The results demonstrated that our spurious contact removal step with default threshold consistently improved the downstream binning outcomes for real datasets (see Supplementary Note 7). Specifically, the MetaCC binning without spurious contact detection only retrieved 75, 101, 6, and 412 near-complete MAGs from the human gut, wastewater, cow rumen, and sheep gut metaHi-C datasets, respectively. With the inclusion of the spurious contact detection step, these numbers were improved to 79, 103, 8, and 417, respectively. Moreover, the total count of high-quality MAGs recovered from the four datasets were also increased from 118, 205, 68, and 696 to 124, 209, 71, and 708, respectively, after the spurious contact removal.

We sincerely thank the reviewer for this important comment for strengthening our results. We have also incorporated the rationale and the limitation of threshold selection for the spurious contact detection step in the **Discussion** section as follows (Page 21 Line 933):

‘In the spurious contact removal step, it is important to note that there is no gold standard for determining the threshold value, as the fraction of spurious inter-species contacts among all Hi-C contacts varied due to the quality of metaHi-C experiments. Moreover, there exists a trade-off in selecting this cut-off value. Opting for larger thresholds can eliminate more spurious contacts but may also result in the unintended removal of a higher number of informative intra-species contacts. Therefore, we have taken a conservative approach by selecting a small yet safe threshold (i.e., the default 5-th percentile) to mitigate the loss of important Hi-C information. From our experiments on the synthetic yeast dataset, the default cut-off enabled the removal of 19.3% of spurious inter-species contacts while incorrectly discarding less than 0.5% of informative intra-species contacts. Furthermore, we conducted experiments to evaluate the impact of the spurious contact removal step using the default threshold on the downstream binning results of all four real metaHi-C datasets. Our results consistently demonstrated that the inclusion of this step with the default threshold led to improved binning outcomes across all datasets (Supplementary Note 7).’

The experiments to evaluate the impact of the spurious contact removal step using the default threshold on the downstream binning results have been shown in Supplementary Note 7 as follows:

'Supplementary Note 7: The spurious contact removal step with default threshold consistently improved the downstream binning results

To assess the impact of the spurious contact detection step using our default threshold on the subsequent binning process, we executed the MetaCC pipeline without the spurious contact removal step. As a result, the MetaCC binning without spurious contact detection retrieved 75, 101, 6, and 412 near-complete MAGs from the human gut, wastewater, cow rumen, and sheep gut metaHi-C datasets, respectively. With the inclusion of the spurious contact detection step, these numbers were improved to 79, 103, 8, and 417, respectively. Moreover, the total count of high-quality MAGs recovered from the four datasets were also increased from 118, 205, 68, and 696 to 124, 209, 71, and 708, respectively, after the spurious contact removal. These results demonstrated a consistent enhancement of the spurious contact removal step in the downstream binning outcomes for real datasets.'

3. Can the full computational time (not just the time for normalization) for the real datasets be provided?

Response: Thanks for this valuable comment, and we have provided the full computational time for real datasets in Subsection 2.7 (Page 11 Line 492).

REVIEWERS' COMMENTS

Reviewer #1 (Remarks to the Author):

The authors have addressed my comments well. I don't have further questions.

Reviewer #2 (Remarks to the Author):

I do not have further comments.